# The opportunities and risks of mobile phones for refugees' experience: A scoping review

**Tiziana Mancini**[1◐]*, **Federica Sibilla**[1◐], **Dimitris Argiropoulos**[1‡], **Michele Rossi**[2‡], **Marina Everri**[3◐]

**1** Department of Humanities, Social Science, and Cultural Industries, University of Parma, Parma Italy, **2** Center of Immigration, Asylum and International Cooperation (CIAC), Parma, Italy, **3** Research and Innovation Centre (NovaUCD), University College Dublin, Dublin, Ireland

◐ These authors contributed equally to this work.
‡ These authors also contributed equally to this work.
* tiziana.mancini@unipr.it

**Data Availability Statement:** All relevant data are within the paper and its Supporting Information files.

**Funding:** This research was supported by CSEIA – Center for Studies in European and International Affairs, University of Parma, Italy (https://

## Abstract

Although mobile phones (MPs) are inexorably changing the forced migration experience, the realm of digital migration studies is still fragmented and lacking an analytical focus. Many research areas are still unexplored, while no narrative, scoping or systematic reviews have been conducted on this topic to date. The present review analyzed scientific contributions in Humanistic and Social Sciences with the aim to provide an overview of existing studies on the role of mobile phones (MPs) on refugees' experience, and to inform practice and policymaking for advancing the use of MPs for the protection of migrants' human rights. A scoping review was conducted using the Arksey and O'Malley framework and the JBI Reviewer's Manual recommendations. A three-step search was carried out in four bibliographic databases by three independent reviewers. Review selection and extraction were performed using an interactive team approach. Forty-three theoretical and empirical contributions were selected, and their content analyzed. The contributions ranged from 2013 to 2018 and varied in terms of disciplines, objectives, methodology, contexts, and migrants' origin, with the most studied group being Syrians. Five different topics concerning refugees' experience and MPs' usage emerged: (a) media practices in refugees' everyday lives; (b) opportunity and risks of MPs during the migration journey; (c) the role of MPs in maintaining and developing social relations; (d) potential of MPs for refugees" self-assertion and self-empowerment; (e) MPs for refugees' health and education. The results showed that modern devices, such as mobile phones, bring both *risks* and *opportunities* for refugees' experience, thereby both favouring and threatening asylum seekers' and refugees' human rights. Recommendations to policymaking and services and associations for advancing the use of MPs for the protection of the rights of migrants have been proposed.

uniprcseia.jimdo.com/). The funders had no role in study design, data collection and analysis, decision to publish, or preparation of the manuscript.

**Competing interests:** The authors have declared that no competing interests exist.

# 1. Introduction

*When you find a place to charge the phone, you see 50 persons around it.*

*[…] In Greece, we slept a night next to the Macedonian borders. There was a man who had a car with his wife; he had an engine from which there was a wire, so we gathered around it. You'd say a spider's web. I stayed 2 days without a phone because of battery. It was dangerous.*

(a Syrian refugee interview [1])

The fields of migration studies, and digital media studies, have consolidated research traditions that have developed separately, and independently, over the decades. However, the increasing ubiquity of Information Communication Technologies (ICTs) in people's everyday lives, including those of migrants leaving their own countries to seek refuge and asylum elsewhere, has progressively driven researchers working in either migration or digital media studies to share their knowledge. Therefore, a new field of research termed *digital migration studies* (for an overview, see [2]) has emerged with the aim to understand the relationship between migration and digital media technologies. Scholars working in this field have focused their attention on how ICTs and mobile phones (MPs), in particular, are transforming refugees' experiences (e.g., [3,4]) prior to, during and after the migration flight (e.g., [5]). New terms have been introduced, including metaphors, to refer to this emerging phenomenon. For instance, the metaphor of the "digital passage" concerns both migrants' routes by the sea (the Mediterranean and the Aegean towards Europe or the Pacific Ocean towards Australia) and the land (the Balkans towards Northern Europe or Mexico towards the USA), as well as the actors that populate those routes. In addition, while the narrative and the visual representation of migrants using MPs have attracted considerable attention in the global public debate, international reports have pointed out that mobile technologies could become essential tools for refugees to claim their rights; such as the right of information and expression, the right to cultural identity maintenance, as well as the right to protection, citizenship and wellbeing in the host country [6]. All this, prompts us to reconsider how refugees' experience should be conceived and studied.

Yet the realm of digital migration studies appears fragmented, unsystematic, and lacking analytical focus (e.g., [7]), especially for what concerns migrants based in Europe or wanting to move to Europe [8,9]. A preliminary search for existing systematic or scoping reviews on the role of MPs in refugees' experience conducted in January 2019 on Scopus and Web of Sciences (core collection) databases did not report any relevant contributions. Reasons for this lack of systematic frameworks derive from several aspects, such as: the unpredictability of a migrant crisis, the emergency and risk conditions during the move, the illegal (and anyway covert) nature of the migration, and the uncertainty of the reception path in the country of arrival. Taken together these aspects have made the refugees' experience hardly accessible to systematic investigations (e.g., [10]). However, it is worth noticing that, differently from the past, digital technologies have allowed for a more direct documentation of the entire migration experience, i.e., from the phase of crisis, displacement and flight to the phases of re/settlement (e.g., [11]).

Building upon these considerations, the present scoping review [12] aims to provide an overview of existing scientific contributions on the MPS role on refugees' experience, and to propose recommendations for advancing the use of MPs for the protection of migrants' human rights [6] contributing to the debate on the relationships between mobile technologies

and human rights in refugees' experience. With the expression human rights we refer to the set of rights sanctioned by the Universal Declaration of Human Rights [13] and to the right of asylum itself, ratified by the United Nations in the Convention relating to the Status of Refugees [14], which is the centerpiece of international refugee protection today and which is grounded in article 14 of the aforementioned Universal Declaration.

## 2. Methodology

The methodology for this scoping review was based on the framework outlined by Arksey and O'Malley [15], on the recommendations made by Levac et al. [16], Munn et al. [12], and by Peters et al. [17]. The structure of this review partially follows that applied by Pham et al. [18] in their scoping review of scoping review study. Specifically, following the recommendations of Levac et al. [16], the review began with the establishment of a research team consisting of individuals with expertise in migration and media studies. For this review, the research team was composed of the five reviewers that signed this work. Through meetings, conducted both live and via Skype, the team defined the broad research question to be addressed and the steps for the study protocol, including the identification of the search terms, the identification of the databases to be searched, the inclusion and exclusion criteria and the methods to solve any disagreement among the reviewers (the protocol can be obtained from the first author upon request).

As suggested by Arsey and O'Malle's [15] framework, the review included the five key phases: (1) identifying the research question, (2) identifying relevant studies, (3) study selection, (4) charting the data, and (5) collating, summarizing, and reporting the results; the optional 'consultation exercise' of the framework was not conducted. Moreover, as recommended da Levac et al. [16], this review: a) used an interactive team approach to selecting and extracting studies, b) incorporated an essential numerical summary and a qualitative analysis of the contributions extracted, and c) identified the implication of the study findings for policy and practice.

### 2.1 Research question

This scoping review was guided by the following question: *Do mobile technologies, especially mobile phones, help refugees cope with the changes and the challenges of their migration experience, therby promoting their security, empowerment, acculturation and well being*? Considering the reasons for conducting a scoping review [12], this scoping review specifically aimed to: 1) map out the body of the published literature in the topic area, in order to give clear indication of the volume of evidences available and an overview of these evidences; 2) to identify key concepts, i.e., to examine reported opportunities and risks of MPs in refugees' experience; 3) to inform practice and policymaking, i.e., to propose recommendations for advancing the use of MPs for the protection of migrants' human rights.

### 2.2 Search strategy

The literature search was conducted separately by three of the Authors of this review (T.M, F.S., M.E.) and it was actioned on January 2019 and ended on January 31th 2019. The first Author (T.M.) conducted the search in three electronic databases: Scopus (with the restriction to Social Science and Psychology areas), Web of Science core collection (with the restriction to Social Sciences areas) and EBSCO database (specifically, Psychological and Behavioral Sciences Collection and PsycINFO). The second Author (F.S.) used two electronic databases: Scopus (with the restriction to Social Science and Arts and Humanities areas) and EBSCO database (specifically, Psychological and Behavioral Sciences Collection, PsycINFO, Social Index). The

**Table 1. Search terms for each concept of the integrative review.**

| Medium | Target | Consequences on refugee's experience |
|---|---|---|
| "mobile phone" OR smartphone OR cellphone OR "cellular phone" OR "cell phone" | "asylum seek* "OR refugee* OR migrant* OR migration OR "unaccompanied minor*" | wellbeing OR risk* OR empowerment OR "migration flight" OR "heritage" OR contact* OR "ethnic network*" OR "social support" OR smuggler* OR "human traffic" OR "traffic network" OR "help relation*" OR acculturation OR adaptation OR citizenship. |

third Author (M.E.) conducted the search in the London School of Economics and Political Science Library Collection Database without any restriction. The databases were selected because they were comprehensive of the topic above described and to covered a broad range of social sciences and humanistic disciplines. No time criteria, language, or type of study was adopted in the database search. The search query consisted of terms that the research team considered useful to describe the general question of the review. Thus, the search terms used to identify appropriate contributions referred to three separate concepts: (a) Medium: i.e., MPs, (b) Target, i.e., forced migrants (asylum seekers and refugees), and (c) Consequences, i.e., the effects of MPs' use throughout the different phases of the refugees' experience; Table 1). The search query was adapted to the specific requirement of each database and, when it was possible, it was applied to the title, abstract, and keywords of each publication.

## 2.3 Study selection

A three-stages screening process was used to assess the relevance of contributions identified in the search.

At the first-stage T.M, F.S., and M.E. imported the citations collected into a single spreadsheet of Microsoft Excel 2011 (Microsoft Corporation, Redmond, WA) reporting the authors, date, title, source, and abstract of each citation into a single line. For this first validation, title and abstract relevance screening were separately conducted by each author that first removed duplicate citations and then excluded citations not eligible for inclusion. At this stage, contributions were considered eligible if they broadly described the role and/or the use of mobile technologies (first of all MPs) in migrants regardless of: the participants' age; the migration type (forced or economic migration) and the phase of the migration experience (crisis, flight, arrival, settlement); the type of contribution (empirical study or other). Papers that did not describe the use of mobile technologies (e.g., contributions that describe the use of the Internet by pc and LAN connections, or studies that used MPs only for collect data in studies not focused on the aim of the review) or that did not focus on migrants were excluded from the analysis.

At the second-stage of the screening process, the collection of 64 non-duplicated contributions (37 from the search made by T.M., 26 from that made by F.S., and 22 from that made by M.E.) remaining from the previous stage was imported into a new spreadsheet of Microsoft Excel 2010 with their Authors, date, title, source, and abstract details. Screening both the titles and the abstracts, T.M. codified the contributions according to three wide criteria: a) type of medium considered (e.g., MPs or ICTs in general); b) type of migration considered (e.g., forced migration or economic migration), c) main topic of the paper. Since the intent was to analyze the impact of MPs on the ways asylum seekers and refugees face the challenges associated with the different migration phases, 16 publications related to voluntary migrants, i.e., economic or circular migrants, such as farm workers, domestic workers or nannies were excluded after the overall agreement of the research team was reached. Because of their focus on the use of ICTs in the refugees' experience in general, two research reports [3,19] and one

editorial [2] were also excluded from the data analyses. Therefore, the corpus of references collected with the overall agreement of the team research consisted of 45 contributions.

At the third-stage of the screening process, the full-test reading of the 45 contributions were attributed to the five reviewers considering their specific area of expertise. Nine publications not directly referred to asylum seekers/refugees or to the use of MPs were identified at this phase. With the overall agreement of the research team, these 9 contributions were excluded from the data analyses. Finally, seven pertinent contributions extracted from the references of the collected articles were selected and, with the overall agreement of the research team, included in the final database allowing a final corpus composed of 43 publications. The flow chart of contributions through identification to final inclusion is represented in Fig 1.

### 2.4 Data extraction

For data extraction, the research team first developed a form to confirm the relevance and to extract characteristics of each full text: it included information about Authors and publication year, disciplinary filed (e.g., psychology, sociology, anthropology, political science), the study's objective, the study's design, migration type and migration phase considered, participants' socio-demographic characteristics, main results collected, and main weakness of the study. Second, the reviewers independently used this form to extract the characteristics of each full text attributed to them.

### 2.5 Data synthesis

The data extracted from the full test reading of each contribution were thus synthetized in a collective spreadsheet (see S1 Appendix). It reports details of the 43 contributions reviewed and includes: a) the type of contribution (theoretical analysis, empirical study, review or meta-analyses) and the methodological details of the empirical studies; b) the migration group considered, including number of participants, migrants' nationality, gender, and country where they actually lived; and c) the emerging topics covered by each contribution (e.g., media practices, effects on social capital, etc. See Results section for details).

To summarize and report the results, first an overview of the body of contributions extracted have been done reporting their general and methodological characteristics, as described in the second and third column of S1. Then, following the five key concepts extracted and described in the last column of S1, a qualitative analysis of the opportunities and risks of MPs in refugees' experience reported in the contributions collected was carried out. Finally, to inform practice and policymaking, a discussion about the consequences of MPs' use for the protection of migrant's human rights were conducted. Prisma ScR Checklist item has been compiled (see S2 Appendix Prima ScR Checklist).

## 3. Results

### 3.1 General and methodological characteristics of included contributions

All contributions, except for one [21], were published between 2013 and 2018. More than a third of the articles (15 out of 43) were published in 2018.

From a methodological point of view, the majority of contributions (37 out of 43) concerned empirical studies based on qualitative research methods such as: interviews (6), focus groups (5), and case studies (2). Sixteen empirical studies used mixed-methods (e.g., interviews and focus groups, or interviews and questionnaires); while five studies used ethnographic methods. It is worth noticing that the qualitative studies were exploratory, and that the recruitment process was often reported as being difficult given the conditions of participants with

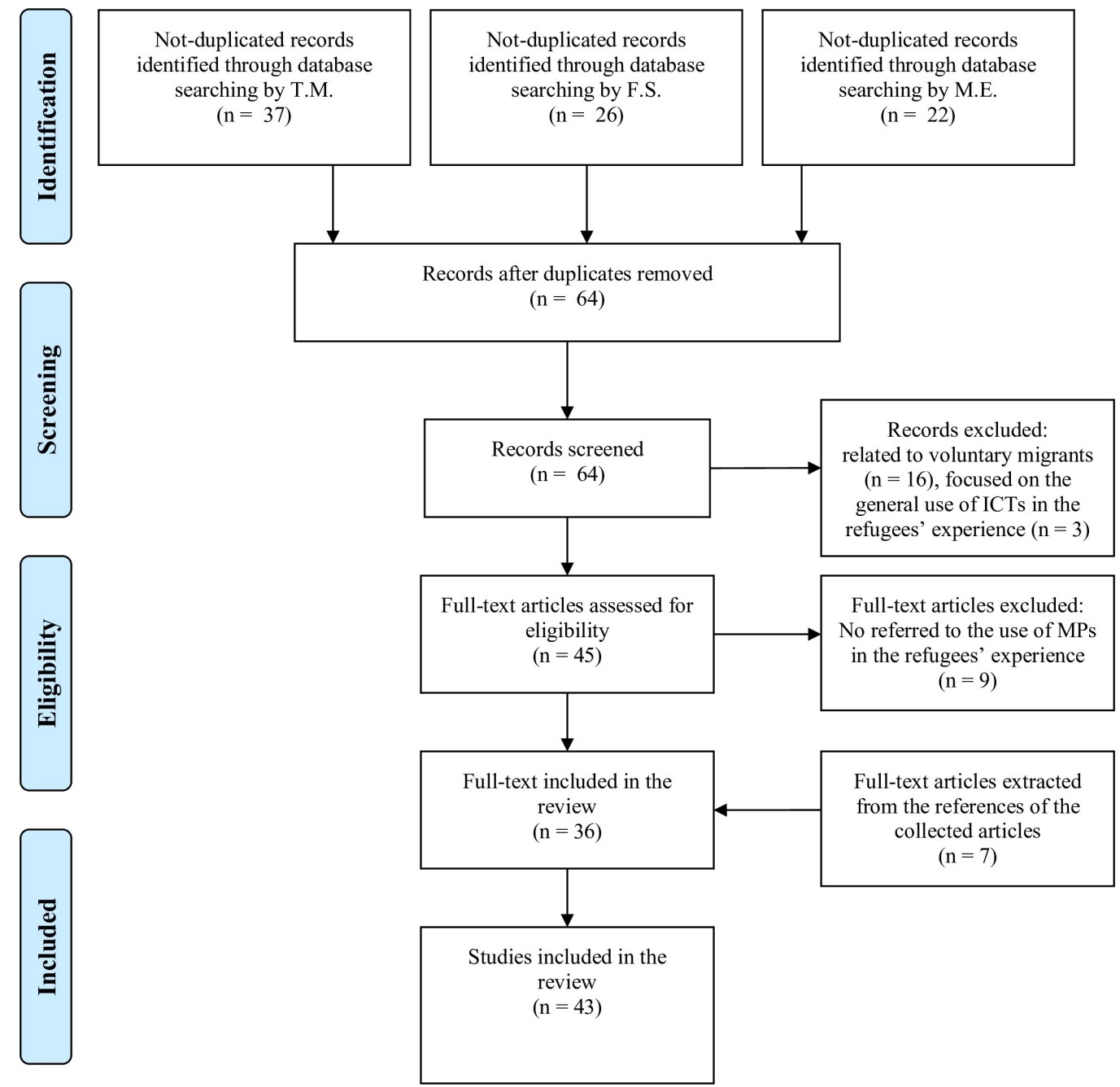

**Fig 1. Flowchart of study selection process (adapted from the PRISMA statement by Moher and colleagues [20]).**

some of them living in refugee camps. Few studies relied on large-scale surveys (e.g., [22–25]). Five contributions were theoretical and carried out an ethical-political analysis of MPs usage in a more general sense (e.g., ethical questions related to the use of big data) during the migration journey (e.g., [26,27]).

All contributions addressed forced migrants (asylum seekers and/or refugees); namely, displaced people from their countries due to persecution, war, violence or threats, who looked for protection in western countries. The most studied group of forced migrants were Syrians: 14

out of 37 empirical studies focused on Syrians, five on Hazara, five on Sudanese, three on Somali, and three on Iraqi refugees. Three studies considered Burmese Chinese descendants, one study on North Korean refugees, one on Tamils, one on Eritreans, and one on Mexicans. Other empirical studies (seven contributions) considered refugees from different origins (e.g., [28–30]) or did not report participants' origins (two contributions). Empirical studies have generally considered refugees living in western countries; namely, Australia (seven studies), USA (two studies), Canada (one study), Germany (four studies), Netherland (two studies) and other European countries (Sweden, France, Italy = one study), as well as South Africa (two studies) and Kenia (one study). Only two studies focused on displaced people, and four studies on people living in refugee camps. The journey of refugees was discussed in three theoretical contributions (e.g., [10]).

## 3.2 Opportunities and risks of MPs in refugees' experience

The key concepts of the collected contributions covered five emerging topics: the first topic concerned the description of *media practices* enacted by forced migrants (10 references). The second topic concerned the *opportunities* and *risks* of MPs' usage in the migration processes (9 references). The third referred to the effects of mobile communication on the *development and maintenance of social relations and networks* (14 references). The fourth focused on the potential and limitations of MPs for *self-assertion and self-empowerment of migrants* (4 references), and the fifth referred to MPs applications for *educational or therapeutic purposes* (6 references). Although there are some overlaps among the different contributions, they have been classified as belonging to one of the five topics looking at the core focus of the study.

**3.2.1 Migrants' media practices in everyday life.** Ten studies converged to emphasise the centrality of mobile technologies in the day-to-day practices of refugees; from reading the news, to being informed about the events in the country of origin, to keeping contacts with family and friends. A majority of the studies showed that one of the main needs of the daily life of refugees was the possibility to access information. Indeed, the condition of instability that refugees experience in accessing news and personal information, namely information precarity, may leave them vulnerable to misinformation, stereotyping and rumors that affect their economic and social capital [31]. As reported in Mansour's study [32] on Syrian refugees displaced to Egypt, there has been limited attention paid to information-seeking behavior, notwithstanding its relevance. Focus groups and in depth-interviews with 37 participants allowed for the exploration of how Syrian refugees accessed information. Results showed that refugees' top priorities concerned: being aware of the situation in their home country, looking for services for their children, shelter and aid in general, together with rights and obligations related to their status such as the right to work in the host country. MPs were the main device used to accomplish these tasks. In addition, more than two/thirds of Syrian refugees showed high *digital literacy* skills, with a preference for social networks; and several applications through which they kept close contacts with aid organizations, besides their families and friends, during and after the migration flight. A larger quantitative study [22] involving 83 Syrian refugees recruited through reception structures in Germany, confirmed these findings and allowed for their generalization. Borkert, Fisher and Yafi [22] found that 95% of their participants used smartphones during their journey to Europe and demonstrated an advanced degree of digital connectivity and digital literacy. More specifically, 89% and 84% of refugees shared information on their journey via WhatsApp, Viber, and similar text or voice message, with 42% and 39% doing so daily. Also, the majority learned their best route through Europe using social networks; none used books or library computers for that purpose. Therefore, refugees emerged as both consumer and producers of digital knowledge through the wide usage of MPs, which

eventually allowed them to make their way to a new life. However, the appropriation and usage of smartphone in a host country can be challenging, especially in the initial phases of resettlement.

Kaufmann [33] provided an in-depth investigation on MPs practices looking at how ten young adult refugees from Syria incorporated these devices in their everyday life practices. Results showed that refugees needed to re-appropriate their devices starting from familiarizing themselves with the information landscape of the host country. Central to this process were four main aspects that defined refugees' everyday practices though MPs in the host country: (a) Geographical orientation and place-making, (b) Language learning and everyday translation, (c) Information access and self-help, (d) Doing family, namely fostering intimate and close relationships. As widely discussed in the next sections, the latter is one of the most common and valued practices supported by MPs. As also reported by Harney's study [30], that focused on three case studies of refugees living and working in Italy, 'the mobile was used as a tool to solve the everyday problems of survival (. . .) it could be used to contact home and friends if need be, so providing a reassurance against the isolation of their migratory conditions' (p. 548).

Alencar, Kondova, and Ribbens [34] reiterated the crucial role of MPs for refugees' everyday life practices, especially during the migration flight. Their qualitative study with 16 male refugees, mostly from Syria, showed that MPs serve different functions; namely, they become companions, serve as organizational hubs, lifelines and diversions. In other words, MPs afford a sense of security, besides allowing migrants to preserve memories of the journey through the storage of pictures taken of important moments experienced during the flight (see contributions reviewed in paragraph 3.2).

Significantly, Wall, Campbell and Janbek [31] point out that MPs can become a double-edged sword, in that their affordability is vulnerable to the availability of connection, the possibility to access alternative SIM cards, and continued government surveillance, especially in the case of the Syrian crisis. Indeed, Wall et al. [31], who based their study on 10 focus groups with between 4 to 12 Syrian refugees in Jordan camps, showed that mobile phones allowed for coping with information precarity in new and creative ways, such as developing a coded language to avoid online surveillance.

The possibility to access information, however, can open the path to new vulnerabilities for migrants, since irrelevant or dangerous information that might threaten refugees' wellbeing can easily circulate in the camp. The study of mobile phone practices in one refugee camp was also at the core of a systematic, large-scale investigation (N = 234), based on observations, interviews, and surveys carried out by Maitland and Xu [23]. Results provided new insights into emerging practices related to MPs usage as a way to cope with the poor quality of the technical infrastructures of the camp. Migrants owned and shared multiple carrier's SIM cards which allowed them constant connectivity via their own MPs. Despite SIM cards storing a unique identity number (phone number), security authentication, personal contacts and other information people typically hesitate to share or to which they want consistent access, SIM-card sharing was found to be a frequent practice in the camp. This was interpreted as a creative way to cope with the challenges of camp living, thereby showing the agency and resilience of refugees. This aspect was also explored in a qualitative and ethnographic study carried out in a refugee camp in Sanliurfa (South-Eastern Turkey) and in a community centre in Istanbul [35]. Nine professionals working with refugees were interviewed; 33 Syrian refugees were involved in informal interviews and almost 60 hours of observations were carried out in the Syrian community centre in Istanbul. Results highlighted the interplay between individual and collective ownership of mobile technologies, which was linked to an informal economy of solidarity among refugees.

Two studies addressed MPs practices focusing, respectively, on Somali and Afghan refugees [21,29]. Despite differences in the social-cultural and political background of these groups as well as their host countries, the media practices reflected those observed with Syrian refugees. Charmarkeh [29] provides an overview of media usage during the migration path and settlement of Somali refugees in France using a critical and multi-sited ethnographic method based on questionnaires, and individual and group interviews. Results confirmed that refugees knew how to use social media and used them, especially Facebook, YouTube, Skype and instant messaging services, on a daily basis both during migration and the resettlement process. Similarly, Glazebrook [21] investigated the usage of MPs through ethnographic observations and 42 interviews with Afghan Hazara migrants resettled in Australia under a temporary visa status. Despite this study being temporarily outdated, it sheds light on mobile communication processes when refugees are under a particular condition in the host country: The possibility of relying on MPs allowed refugees not only to stay in contact with families and friends during critical conditions, thanks to free call services during the night offered by phone providers in Australia, but also and more importantly, they could collect information on how to self-administer their everyday lives under a temporary status visa (e.g., applying for a government health card; finding and leasing a house; understanding changes in visa policy and obligations; applying for and keeping a driving license; financing and insuring a car; opening a bank account, etc.).

**3.2.2 Media practices during the migration flight: Risks and opportunities.** A more detailed discussion of MPs opportunities and risks was found in nine articles that moved from theoretical analyses to empirical analyses based on both big data, unofficial sources; and refugees' self-reports focused on the role of mobile technologies during refugee journeys. They addressed MPs usage "on the move"; namely, while migrants were moving to a safer place, between and across the borders. These articles were centred on the recent Syrian refugee crisis and showed that Syrian refugees faced the same challenges as other migrants who, in the past, were forced to leave their home countries. However, the Syrian refugee experience differs in at least one way: their route to safer places did not only depend on physical barriers, but also on the access and availability of mobile technologies (e.g., [3,4]).

Starting from a theoretical reflection on the role of big data in transforming research on forced migration, three contributions stressed the need to define this new research area, which seems to be characterized by a profound ambivalence. Curry, Croitoru, Crooks, and Stefanidis's [26] analysis highlighted that MPs are widely used among migrants; therefore, a new generation of sources, data and information have become available for research purposes. Crossing big data, official sources and maps, Curry et al. stressed this innovative research space as one that has a high potential; however, it is dense in ethical, political and methodological implications. As Latonero and Kift's [27] theoretical analysis pointed out, the study of the refugees' "digital passage" should be subjected to an ethical-epistemological status that guarantees privacy and sensitive data protection, namely the refugees' right to privacy and security. Similarly, Beduschi's [10] theoretical analysis emphasized the limits and the risks posed by the unrestrained use of new technologies. However, he highlighted how the study of big data and the access to private information can shed light on covert phenomena, such as: human-trafficking, unpublished agreements between states, and illegal refoulements.

Three empirical contributions analyzed digital practices during the refugees' journey ([1], [36], and [37]). Findings on Syrian, Iraqi and Mexican refugees revealed that MPs were widely used prior to, and during, the migrants' journey, constituting an essential part of it. Distances, roads, altitudes, temperature and weather conditions, currencies, languages and possible shelters or refueling points were processed through MPs applications. For some migrants, these practices were similar to those of business trips and holidays. According to several

interviewees, the preparation of the trip required giving particular attention to the correct functioning of the phone, such as bringing battery chargers and plastic bags to keep the mobile dry. Some migrants highlighted how the mobile allowed them to avoid carrying many objects such as torches, maps, cash money, dictionaries and even documents. An interviewee described the hasty escape preparations as an attempt to digitize and store in the mobile as much as possible of his previous life, in an attempt to preserve it. Many considered their mobile phone as a "travelling light" which was important to escape controls in starting nations. Refugees' ability to send details of their location to coastguards, friends, or family members while on the move was a matter of necessity and enabled the "distant proximities" that make life bearable [38]. Getting information or being guided from far away while crossing rivers, seas and mountains was defined as "salvific" in the study by Dekker and coll. [37]. The prospect of losing or damaging their own mobile phone raised a deep existential and physical insecurity in refugees interviewed by Gillespie and coll. [1].

Having access to electricity to recharge the phone, and access to the Internet, often became a question of life or death in the refugee journey. Most of the respondents reported that they continually "swap, change and share batteries". Refugees risked being disconnected without battery and remaining without connection even for a short time meant failing to meet or deliver money to a smuggler on time, getting lost, or being separated from companions. Among the main advantages offered by MPs, refugees stressed the possibility of signaling dangers encountered in the sea, verifying through social networks paths and passages, organizing together with other refugees and with locals the crossing of the borders in an attempt to elude and overcome controls, and informing loved ones of their safe arrival. However, mobile connectivity can imply risks and exploitation: many refugees reported that they became victims of fraud. Without an identity documentation (e.g., evidence of home address on a utility bill) it is not possible to register with a mobile network. Refugees often asked local people to buy a SIM card on their behalf. This was in some cases risky and unreliable—and illegal—since it made refugees vulnerable to blackmail or exploitation [1]. In some circumstances, e.g. when passages to Europe were interrupted and migrants had to spend time in refugee camps, MPs became a sort of currency. They were bought and sold, exchanged and bartered for, fought over and gifted.

In line with the studies previously reviewed (e.g., [35]), some contributions showed that MPs were also co-used by entire families or social groups traveling together. Many respondents traveled in groups, and only few members of these groups were in charge of planning and navigating using their MPs [37]. Moving in groups was a common strategy to share resources but also to address the many dangers of digital migration [27]. Indeed, the literature highlights several risks associated with the MPs' use. As illustrated by Beduschi [10], through the use of GPS applications migrants can be tracked and discovered by border guards, traffickers, smugglers, and common criminals. The study by Gillespie et al. [4] pointed out that using MPs is at once a lifeline and a risk, making specific survival strategies necessary. Syrian refugees, due to the repression faced in their country, replicated particular subversive MPs practices during their journeys. For example, many of them protected their digital identities, and any information about intended routes and destinations, using closed Facebook groups and encrypted platforms, such as WhatsApp, to connect with smugglers and others on the move. Moreover, many of them used avatars and pseudonyms on Facebook to avoid online surveillance by state actors in Syria or by other hostile groups. Similarly, Newell and colleagues' [36] study on Mexican migrants observed that at the border, the use of phones was a double-edged sword: the disclosure of phone number contacts was the object of extortion and abuse by thieves, human traffickers, drug traffickers, and even corrupt police officers; therefore, having a list of phone numbers became a risk.

Gillespie and colleagues [1] argued that beside the possibility to document and share personal stories, to gather new information and to co-produce knowledge, migrants' usage of MPs is associated with the risk of enabling the circulation of misinformation and exposing them to the risk of untold harms and even death; for instance, when recorded images of torture or abuse fell into the hands of the perpetrators. Along African routes, smugglers and traffickers used migrants' personal MPs to blackmail families of kidnapped migrants, who were forced to listen on-line to the screams of their relatives subjected to torture to pay for their release. Many accounts revealed that once migrants were in the hand of traffickers, the first thing that happened to them was the deprivation of their mobile phone. Chouliaraki and Musarò's [39] case study on two of the main border sites of the 2015 migration "crisis", Italy and Greece, observed how police and national authorities used communication technologies to identify non-authorized migrants before they reached national borders. That is one of the reasons why many refugees mistrusted digital help services even when provided by NGO websites or chats and preferred informal channels and networks [37].

Lastly, two articles consider the potential of digital technology, especially social media networks, as 'digital witnesses' of the sufferings documented by refugees and shared by them on platforms such as Facebook. In their study, Rae, Holman and Nethery [40] analyzed how social media, accessed primarily using MPs, became a vital tool for asylum seekers within Australian offshore detention facilities to connect with journalists, advocates, activists, legal representatives and family and friends. Through an analysis of two Facebook pages accessible to the general public–related to the case of a Kurdish journalist and Iranian national, Behrouz Boochani, and to the case 'free the children NAURU' dedicated to asylum seekers' children detained in Nauru–Rae and colleagues showed that detainees circumvented the usual mediation of their stories and engaged in a self-represented witnessing. Forced migrants challenged public and political debates representing themselves with their own voice using online platforms to share their stories about the flight and their experiences of migration detention. Yet, the authors concluded that the message conveyed through self-represented witnessing is likely to be limited to the audience of social media networks. In fact, Boochani and 'Free the Children NAURU' received a good number of followers; nevertheless, it was only when the case was picked up and reproduced by mainstream media that the message was able to reach a broader audience. Risam [41] explored further the relation between media produced by refugees and mainstream media: looking at 20 United States and United Kingdom newspapers he selected articles on migrant-related selfies in the context of the Syrian refugee crisis and analyzed them through a quantitative textual analysis. The author concluded that refugees' selfies that circulated among the public did not convey the original contents. Instead, the lens of the photographers captured the act of selfie-taking and framed the image for the public abstracting them from the contents of the original selfies. Moreover, beside these "migrant-related selfies", the content of the articles often reflected suspicion, lack of sympathy and, at times, antipathy for migrants.

**3.2.3 MPs for the maintenance and development of social relations.** If keeping in touch with loved ones left behind, or with other refugees, resulted in one of the main reasons for using MPs in refugees' experience (see paragraph 3.1), it is not surprising that fourteen contributions relied on the role of mobile technologies in the development or maintenance of social relations before, during and after the journey. With the exception of two contributions that respectively focused on youth migrants stranded in Addis Ababa waiting for visa clearance to be reunited with parents [42] and on Sri Lanka Tamil residents of a refugees camp in India [43], ten out of fourteen contributions concentrated on the resettlement of refugees in the new society. Only two publications, referring to the same study, covered the entire migration path, and in this case referred to a sample of unaccompanied foreign minors [28,44]. All articles

explored the role of communication via MPs' in sustaining migrants' transnational social net-works–thereby maintaining their bounding social capital (characterized by strong ties with family members and close friends [45,46])–or/and encouraging migrants' participation in the host society–thereby, enhancing their bringing social capital (characterized by weak links among members of a heterogeneous network [47]). However, the conclusions to which the studies arrived differed in the role of MPs in facilitating refugees' adaptation to their actual living conditions and on sustaining their wellbeing.

The ethnographic study with 29 refugees living in South Africa by Bacishoga, Hooper, and Johnston et al. [48] reported that MPs played an important role in the development of bringing social capital, thus supporting social and economic integration of refugees into the new community. Indeed, MPs favored weak links with people with whom refugees studied, worked, and shared accommodations or similar interests, also aiding refugees to get involved in different group activities. In addition, MPs contributed to maintaining and strengthening the existing strong ties with family members and homeland national friends and relatives, which helped refugees to overcome isolation and to feel accepted, confident and safe, while not facilitating the development of a new bounding social capital. Wollersheim and collaborators [49–52] found similar results from a wide intervention program dedicated to 111 women from different refugee backgrounds living in Melbourne (Australia). The program consisted of providing women with five-week face-to-face peer support training and with unlimited free-call fixed-dial mobile phones for one year. Authors collected data derived from this program using different methods: two focus groups with nine participants from Nuer background at the end of the 5-week-block of peer support training program [49]; interviews conducted at the end of the program with twenty-nine women [51]; administration of pre- and post-intervention questionnaires [50]; quantitative analyses of the logs of outgoing phone calls [52]. The results are reported in different publications; they coherently evidenced both an increase of existing social capital (bounding social capital) and a creation of new social capital (bridging social capital). As for bounding social capital, refugees reported that the program provided most of them with the opportunity to meet other women in their community to create stronger relationships among peer support group members, and to strengthen ties with family and friends left behind in their home country. Bounding social capital improved women's personal empowerment (e.g., having access to new technologies, providing more opportunities to learn and practise English) and enhanced their wellbeing (e.g., increased their sense of happiness, reduced social isolation). To a lesser extent, however, MPs facilitated women's interaction with Australians and with government institutions; for example, helping refugees to overcome English language difficulties, and to acquire host society cultural norms and behaviors useful to daily life in Australia.

The role of MPs on refugees' adaptation and wellbeing in the new society were also considered by some studies. Analyzing five focus groups conducted with 29 Syrian youth resettled in Ottawa (Canada), Veronis, Tabler, and Ahmed [53] found that social media facilitated the agency and the sense of control of participants in their process of resettlement and adaptation to the new society. Findings suggested that MPs, besides helping migrants to access information they needed and to learn about the new society, provided a "virtual contact zone" where the youth built transcultural connections, i.e. they bridged cultural differences and negotiated Syrian and Canadian cultures.

In a study that explored the media selected by 29 Hazara male youths in the resettlement process in Brisbane (Australia), Tudsri and Hebbani [54] found that Internet-supported communication helped migrants to integrate into the host society and to become bicultural individuals [55]. Integration, however, depended on linguistic concerns: only migrants who had limited English language skills, but were motivated to advance their English proficiency,

consciously used MPs to become "more Australian" (i.e., to make contact with others in Australia), while still maintaining contacts back home. Instead, youths with similar English proficiency, but with no interest to advance it, used media only to contact their parents in Afghanistan and to keep in touch with their original culture and language, thus showing a tendency towards separation [54]. A trend towards separation was also highlighted in the interviews that Kang, Ling, and Chib [56] conducted with 20 North Korean women settled in South Korea. The results showed that MPs indirectly facilitated women's withdrawal from South Korean society. To bypass the social barriers and social stigma, refugees strategically avoided direct interaction with South Koreans and hid their identity and accent; for example, seeking help online anonymously, using only text-based communications, and manipulating their identity on social media.

Besides language skills [54] and stigmatization of refugees [56], other barriers drove the use of MPs limiting the integration of forced migrants into the new society. The digital divide was a recurrent topic in many reviewed contributions. For example, Kutscher and Kreß's [28, 44] study outlined the unequal distribution of digital opportunities among unaccompanied minors that arrived in Germany in 2015. The digital divide derived from limited media literacy skills and structural conditions (e.g., internet access, device availability, professional assistance in Germany youth welfare institutions), which prevented the use of Internet connectivity before, during, and after the minors' flight. Kutscher and Kreß's study, however, stressed the relevance of MPs during and after the migrants' journey. In line with the studies on MPs risks and opportunities described in paragraph 3.2 (e.g., [1], [36], [37]), Kutscher and Kreß's results showed that MPs assisted young migrants during the flight, helping them to get in touch with facilitators, navigate through unknown areas, receive and share information, and make emergency calls. MPs facilitated the linguistic integration of minors into German society, thanks to the large number of applications available for language learning and translation. Moreover, Facebook, WhatsApp, Viber, and Skype played a vital role in maintaining migrants bonding with their social capital after they arrived in Germany. Social media were mainly used to stay in contact with family and friends and/or with peer refugees met during the flight. Alam and Imran [57] found similar disparities in the physical access and use of digital technologies among 28 refugees from diverse ethnic backgrounds. Also in their study the digital divide–related to the migrants' digital skills and to the ability to pay for the services–was a critical aspect of refugees' social inclusion in the broader Australian community. Similarly, cost, connectivity, and perceived usefulness, influenced MPs use in the 29 South Africa refugees that participated in the ethnographic study by Bacishoga, Hooper, and Johnston [48].

Structural barriers to connectivity and the digital divide can also limit the access to online support. Mikal and Woodfield [58] investigated online communities of support to understand the extent to which these communities could buffer refugees' post-migration stressors. The content analysis of the focus groups conducted with 25 refugees living in the USA, and coming from Iraq and Sudan, showed that participants were reluctant to look for online communities of support. The hesitancy was due to both cultural differences and to barriers to the use of mobile technologies, including media literacy, safety concerns, and limited connectivity. Results of this study showed very limited evidence on refugees' usage of MPs as a tool for reducing post-migration stressors. Sreenivasan and colleagues [43] found similar results. Mobiles allowed Sri Lanka Tamil refugees, residents of the Mandapam camp (India), to create a virtual community, which migrants considered important in strengthening ties with family and friends in Afghanistan, as well as in fostering personal and professional ties among Tamil refugees in the camp. However, telecommunications did not have a reparative function from the trauma experienced by some refugees. On the contrary, the usage of MPs emphasized their sense of insecurity and fear of being surveilled. Similarly, the positive feelings associated with

online communication with loved ones abroad did not fully compensate the emotional land-scape of the everyday life of young Somali migrants stranded in Addis Ababa. Transnational communication fostered affective capital, one of the only sources of capital young migrants can have. The affective capital allowed young migrants to manage anxiety and provided them with a feeling of trust; however, this increased migrant dependency, being online connections depending on remittances they received from family members abroad [42].

**3.2.4 MPs and migrants' self-empowerment and self-assertion.** Adopting a different point of view, four contributions highlighted that mobile technologies can support forced migrants' individual empowerment, contributing to their self-affirmation. In a study con-ducted with 176 asylum seekers in Germany coming from 20 different countries, Witteborn [25] focused on how migrants used mobile technologies to manage how they presented them-selves to others. Results showed that the medium allowed migrants to oscillate between being perceptible and being imperceptible, and that depended on the specific relational context. For example, migrants tended to use mobile technologies to become imperceptible in terms of "asylum seekers" or "refugees"–labels that were often considered as undesirable and that they tended not to identify with–and perceptible in a more desirable terms, e.g. proposing past selves or ideal selves. Furthermore, allowing messaging, calls or video calls, online communica-tion allowed migrants to become perceptible by people located in geographically distant con-texts–typically family members left in the home country. This fostered co-presence and the reconstitution of family intimacy, thereby positively affecting their personal well-being. In the interactions with family members and relatives, migrants tried to manage their self-presenta-tion stressing the positive aspects of their life away from home rather than their everyday life difficulties. In addition, in line with Rae et al. [40], Witteborn [25] found that mobile technolo-gies allowed migrants to present to society at large by overcoming their perceived invisibility and making themselves visible as a political force that comes up with ideas, and mobilizes and claims their own rights.

A political usage of MPs also emerged from the study by Rohde and colleagues [59]. Through the analysis of interviews of Syrian people who were FSA fighters, activists, or refu-gees during the Syrian civil war, the authors found that at the time of the civil war, phones and connectivity were often inaccessible to people, and that there was a strong, shared fear of being surveilled by the government through technologies. Syrian rebels developed different strategies to elude government surveillance and MPs had a central role in this; for instance, filming, reproducing and sharing videos were possible both when the Internet was accessible and when it was not thanks to MPs technology (e.g., via Bluetooth). Such videos could be professional or amateur, but usually had political and social content: they recorded atrocities, such as killings and mistreatments, that occurred during the war, and were used for denunciation or political propaganda purposes. Facebook was used as a tool for political and social commitment as well, allowing the organization of collective actions and the dissemination of information.

The possibility offered by mobile technologies for refugees' self-assertion has been highlighted by Leurs' study [60]. Leurs considered the use of MPs for the support of funda-mental rights–e.g., the right to information and the right of expression–investigating the expe-riences of 16 young Syrian refugees in the Netherlands. The author found that MPs allowed the young respondents to exercise the right of expression: migrants used various digital plat-forms such as blogs and social networks to communicate, report, and do politics avoiding cen-sorship through different strategies. In line with the study presented in paragraph 3.1 (e.g., [22], [31]), results showed that mobile media facilitated the free circulation of information, including that provided by people online, thereby enforcing the right to information. MPs rep-resented a window on the world as well: respondents used MPs to see how in other countries people lived in a state of peace; this made them aware of their state of insecurity and

precariousness because of the war. This awareness prompted migrants' self-determination; for instance, Karim (one of Leurs's study participants) moved by this awareness decided to leave Syria in an attempt to regain control of his life.

Twigt's [61] interviews with 52 Iraqi refugees in Jordan highlighted that transnational digital connections afforded by mobile technologies is literally "vital", especially for coping with the hardships of migration such as waiting for a legal status. As also pointed out by other studies (e.g., [42]), MPs can become both a vehicle of hope, which is reinforced by transnational digital connections through the sense of being close and physically together, and a vehicle of despair since they can notify deaths, wars and loss reports of countrymen, including acquaintances.

**3.2.5 MPs' usage for therapeutic and educational purposes.** The usage of MPs for therapeutic and educational purposes is another topic which emerged from the literature analysis. Three contributions focused on the use of MPs to assess psychopathology and provide therapy to forced migrants, indirectly stressing the role of MPs for migrants' rights to health. It is well known how factors related to the pre-flight (e.g., war-related stressors such as torture and rape) and to the post-flight period (e.g., cultural integration issues, loss of family and community support) make refugees a vulnerable population for the development of psychopathology (for a review, see [11]). Nevertheless, administrative and practical obstacles, linguistic barriers, and the limited availability of mental health professionals, often limit the access of refugees to mental health services [62]. A solution to these problems can be found in e-mental health interventions that are typically shorter than the traditional ones, partially automated, and can be used totally or partially offline, in situations of limited connectivity. E-mental health interventions have mainly been carried out through websites; more recently the dissemination of MPs Apps has allowed for a widespread access by refugees and asylum seekers. Sandoval, Torous, and Keshavan [63] stressed the advantages of these applications, such as: affordability, accessibility, minimal commitment, engagement, and lack of stigma, and discussed the clinical case of Mr A., a young Eritrean who lived in a refugee camp for 9 months before arriving in the United States. Once he arrived in the US, he started experiencing psychiatric symptoms, received a diagnosis of schizophrenia, and started following traditional psychiatric treatment. However, Mr. A. felt the need for additional help. Using his mobile phone, Mr. A. found e-mental health Apps which eventually helped him to monitor his symptoms, to learn and practise some useful skills, to be educated about his condition, and to practise in scenarios relevant to his recovery plan. In other words, the Apps allowed him to develop some skills that face-to-face therapy did not allow for. In the long term, Mr A.'s usage of Apps became an expansion and integration of the face-to-face therapy.

MPs can be used also for screening purposes. Tomita, Kandolo, Susser, and Burns [24] explored the use of short messaging services (SMS) for screening depression within refugees. In particular, they analyzed a refugee population in the city of Durban, South Africa and used a longitudinal design consisting of baseline (N = 153) and follow-up (N = 135) assessments. Comparing SMS and face-to-face methods, results confirmed the viability of the SMS-based method to screen for risk of depression among refugees in low-resource settings. Face-to-face and SMS-based methods resulted as equally able to screen for depression, while no significant difference in the assessment method preference and a higher comfort of use of the SMS-based method emerged. Taken together, the few studies related to the use of e-mental health programs/Apps seem to report encouraging results. Nevertheless, evidence-based data on the efficacy of these treatments are still limited, and issues about data storage, privacy, costs, problems related to the shared use of the MPs [63] as well as their usability and interoperability [62] have not been addressed and normed yet.

Mobile technologies can have not only therapeutic but also educational purposes. The study by Bradley, Lindström, and Hashemi [64] explained how mobile apps could be used in the first phases of the resettlement in the host country. Their study involved 38 Arabic speaking migrants newly arrived in Sweden and participating in an introduction program to learn Swedish language and culture. The sample was then divided into a control group (N = 14), which only attended the introduction program, and an experimental group (N = 24), which also attended a practical pronunciation course carried out through a mobile app. The experimental group showed better speech flow and intonation, self-confidence in speaking, high satisfaction in using the app, and desire to use the app more often. This study showed that MPs can foster learning and skills development that allow for a faster integration of migrants in the host country. In O'Mara and Harris' [65] study, 24 young migrants with different backgrounds (Vietnamese, Samoan, Sudanese, and Chinese) and living in Melbourne, Australia, were involved in a site-based arts pedagogy program in which various sites and apps were used to create artistic products. Digital media technologies resulted to be excellent for creative and identity experimentation, self-expression, learning, and interpersonal interaction and collaboration among young people from different cultural and linguistic backgrounds, contrasting the negative representation of the use of digital media by young people. The authors concluded that online pedagogies can be used for bridging cultural, gender and educational gaps; in particular, this can happen if MPs are used in the context of educational practices with the aim to experiment, learn, socialize, and grow. Dahya and Dryden-Peterson's [66] study obtained similar results. The study involved 21 Somali women refugees and 248 in the diaspora with the aim to explore mobile technologies usage for higher education. Refugee women are often burdened by social expectations and inequalities that limit them in pursuing higher education. The usage of social networks allowed refugee women to engage in transnational conversations with compatriot peers studying in higher education; this allowed them to gather information about post-secondary schools. Furthermore, these exchanges through social networks allowed for a progressive change of the social and educational landscape of higher education for Somali women that either were refugees or who remained in Somalia. This contributed to creating new opportunities for refugee women, especially to self-determine their own lives.

## 4. Discussion and conclusion

The present work aimed to leverage insights on the role of mobile technologies in refugees' experience. Theoretical and empirical contributions from the emerging field of digital migration studies were selected and analyzed following the recommendations for scoping review. The result is a state-of-the art overview on how mobile phones can favour and threaten refugees' experience.

### 4.1 Overview of literature on the use of MPs in refugee experience

The body of the literature highlights a broad and varied overview of the topic analysed; the contributions spaced across many different disciplines of the humanities and the social sciences. Beyond this disciplinary variety, for obvious reasons, the contributions are for the most part very recent and mostly focused on the Syrians and on the Syrian crisis, and related to the resettlement phase of the refugees' experience. The key concepts of the collected contributions accounted for five different topics concerning refugees and MPs usage. Taken together, the five topics emerged from the literature reviewed can be seen as organised along a continuum (Fig 2) where at one pole of the continuum the focus is on how media–i.e., mobile technologies, especially mobile phones–are incorporated in migrants' everyday life practices in the host countries, or during the journey, or in refugee camps. The articles focused on the first topic

**Opportunities and risks of mobile phones for refugees' experience**

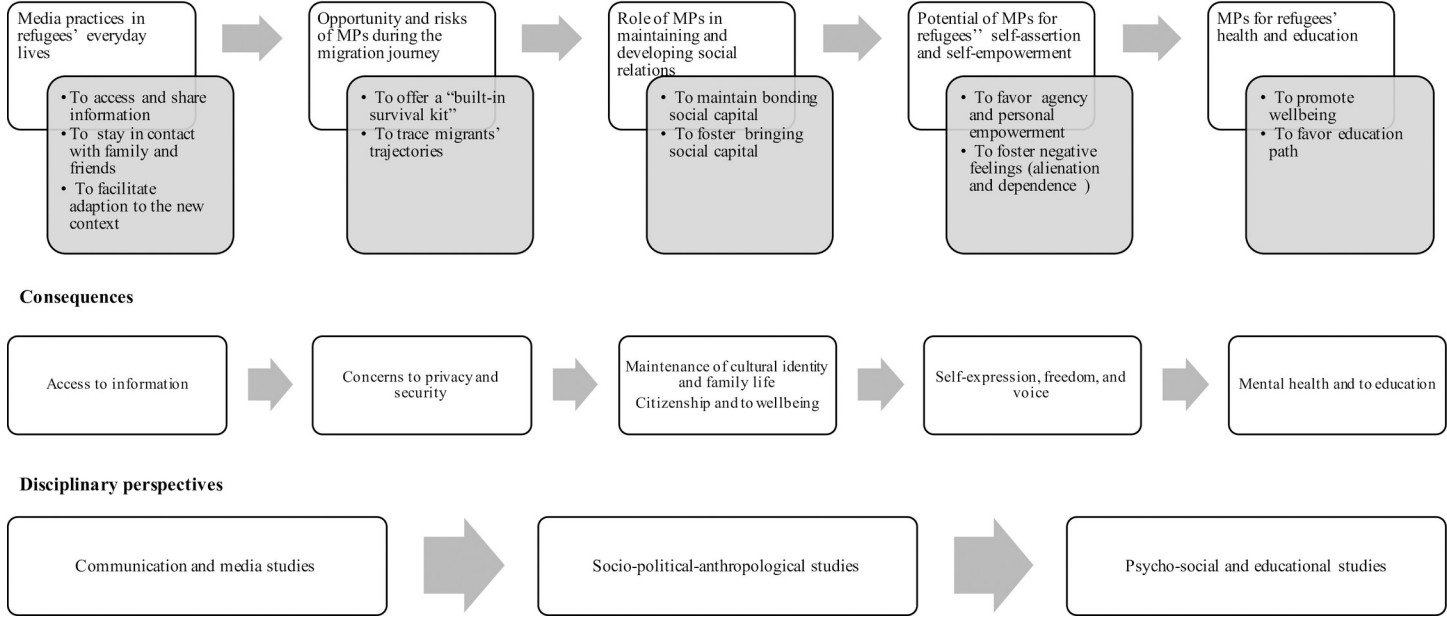

**Fig 2. Overview of literature on the use of MPs in refugee experience.**

highlighting the positive aspects associated with the usage of mobile technologies, such as accessing and sharing information, staying in contact with family and friends, and facilitating the adaption to the new context, thereby highlighting that MPs' can support migrants' in finding information uselful for their experience. Focusing on the opportunities and risks of media practices during migrants' journey, the contributions related to the second topic clearly highlighted that MPs inexorably accompanied migratory journeys, tracing their digital trajectories. This, on the one hand, allows for a more realistic knowledge of the migration phenomenon, but, on the other hand, raises concerns related to privacy and security of migrants.

In the middle of the continous can be located the majority of the articles reviewed here which agree in recognizing the importance of mobile phones for both maintaining contacts with the context of origin (bonding social capital) and fostering new relationships in the host context (bringing social capital; third topic). Therefore, these studies showed that MPs can foster the migrants' maintainance of cultural identity and family life, their sense of citizenship and their wellbeing.

At the other pole of the continuum, MPs are framed as facilitators for migrants' agency and voice, and for the solution of mental health problems or educational needs. Studies showed that in the hardships of the migrants' condition, mobile media affordances impacted on refugees' emotional dimensions, favoring their agency and their sense of personal empowerment, therefore, enforcing their voice. Nevertheless, MPs sometimes also fostered migrants' negative feelings, such as alienation and dependence on the remittances necessary to maintain online connections, thereby threatening migrants' freedom and liberty. Providing examples on the ways in which mobile technologies can be used to promote wellbeing and education, studies focused on new MPs applications for migrants' health and education finally recalled the role of MPs on favoring the refugees' mental health and education and, more broadly, the integration of refugees in the new society.

The transition from one pole of the continuum to the other entailed a change in the disciplinary perspectives of the surveyed contributions: from the perspective of communication and media studies, and the perspective of socio-political-anthropological studies, to the perspective of psycho-social and educational studies. Overall, the articles reviewed here contributed to define a relatively new transdisciplinary field of inquiry [7]. Combining media and migration studies, this emerging field–i.e., the *digital migration studies* [2]–tried to reply to the main question of this scoping review: *Does mobile technologies, especially mobile phones, help refugees coping with the changes and the challenges of their migration experience, thereby promoting their security, empowerment, acculturation and well being*?

## 4.2 The opportunities and risks of mobile phones for refugees' human rights

The empirical and theoretical analyses reviewed across the five topics consistently showed the double-sided role of MPs in the refugee experience. This means that modern devices, such as mobile phones, bring both *risks* and *opportunities* migrants; therefore, MPs seem both favouring and threatening asylum seekers' and refugees' human rights.

Besides calling and texting, the "polymedia" affordances of devices for online communication, such as mobile phones, offer migrants a sort of "built-in survival kit" [1]. This kit includes several possibilities, among them: to make contacts abroad, to seek help, to find better routes, to stay informed about the situation during the journey and the destination countries, to manage risks and opportunities during the journey and when crossing borders. The kit includes also the possibilities to shed light on covert phenomena and to voice the migrants' experiences favoring the agency and the sense of personal empowerment. Moreover, mobile phones allow migrants to keep in touch with home and, although to a lesser extent, to get in contact with services and institutions in the new countries, to learn new languages and, more broadly to improve their knowledge and skills in order to integrate in the new context. Therefore, mobile phones seem to guarantee some fundamental human rights: Among them, the literature acknowledged the right of information and expression, the right to cultural identity maintenance, the right to the family life, the right to mental health and the right to work and education. It is important here to stress that the recognition of human rights has not only been confined to asylum seekers or refugees. From the literature reviewed, the usage of MPs in refugees' experience did not appear as episodic; rather it accompanied migrants throughout the different phases and contexts of migration, thereby creating transcultural communication. Therefore, mobile phones seem to create opportunities to strengthen and develop digital, linguistic, social and educational skills both for migrants and the people left behind in the countries of origin. Moreover, it is arguable that mobile devices promoted social inclusion and the wellbeing of migrants only to the extent that they have allowed the creation or maintenance of a double bond: one with one's own cultural heritage and one with the new society. This double bond is fostered by mobile technologies' affordances through the development of a virtual contact zone where refugees nurtured transcultural connections. Therefore, the use of mobile phones in refugees' experience seem to be a driving force for the increase of the transcultural social capital and the identity capital.

Nevertheless, mobile communication could also be a ubiquitous digital threat; through the same tools, refugees could become victims of human traffickers, tracked and controlled by the regimes from which they flee, intercepted or even rejected by the digital control systems of the countries to which they are directed and exploited and manipulated by mainstream media. Therefore, in the absence of clear ethical regulations, the traces left behind by mobile devices could threat migrants' right to life and security and, consequently, the right to asylum, to citizenship and to wellbeing.

Overall, the literature on the role of mobile phones in refugees' experience is still substantially fragmented and not consistent; therefore, making it impossibile for this scoping review to reach definitive conclusions. The ambivalence of digital technologies in peoples' lives was found in other research strands, such as those concerned with children, youth (e.g. [67]) and family communication [68,69]. Smartphones, in particular, have contributed to the creation of cultural practices that, on the one hand, can expand offline practices, such as contacting friends, finding partners, dealing with organizational issues through application support but, on the other hand, can expose users to different risks, such as overuse or addiction [70] information flaws and other privacy issues (for an overview: [71,72]). However, as pointed out by the studies reviewed here, the positive or the negative role of connectivity–made possible by mobile devices–depended mainly on individual factors (e.g., digital and linguistic skills, motivation to integrate in the society), and on technological (e.g., connectivity, device availability) and social barriers (e.g., stigma and prejudices). Individual, technological and social barriers fed the *digital divide* among refugees, putting women and older migrants at a greater risk. Digital divide affects refugees' acculturation processes in that: the creation of a "virtual transcultural space" can be turned into the crystallization of a "virtual ghettoized space" that did not favor migrants' wellbeing, thereby sometimes contributing to "freeze" their traumatic experiences.

## 4.3 Recommendations

The findings of the surveyed studies can provide useful insights to policymakers and to services and associations operating in different communities, therefore, advancing the use of mobile technologies and, specifically, of mobile phones for the protection of migrants' human rights. In particular, policymaking should consider mobile technologies as a new and open space that requires to be taken into consideration in the application of the right to protection and asylum. Services and associations should consider the usage of mobile technologies as a means to strengthen the personal empowerment of asylum applicants and refugees and/or to implement social and educational programs aimed at fostering or strengthening migrants' personal agency. Specifically, mobile technologies should be considered as a new and open space for redefining the right to protection, to support and to integration of migrants in the new society and, therefore, also as a useful medium for redefining services dedicated to migrants. Being typically designed from the point of view of the receiving societies, services for asylum seekers and refugees often show cultural, social and community interaction barriers. The usage of programs, or applications, via mobile phones could overcome the strongly asymmetrical relationship between migrants and host society in health, education or employment services, as well as reduce the limitations of generalist services. These services present limitations linked to costs, language proficiency, and clarity of rules concerning the interactions between migrants and host society services. Therefore, the usage of programs and applications via mobile phones could foster and strengthen the social and emotional bonds identified as the major determinants of refugees' social inclusion, health and education. Consistently, building upon Glazebrook's thoughts [21], a new generation of services and programs, able to overcome social isolation, linguistic and cultural gaps, could be though. These services and programs need to be built not for, but with forced migrants.

## 4.4 Limitations and indications for future efforts

This scoping review has the merit of having mapped the still fragmented and not consistent literature on the topic, identifying some gaps in the literature. It certainly constitutes a precursor

to systematic reviews on more specific topics, such as those concerning the role of MPs in the Syrian crisis. Despite this, the present scoping review has some limitations.

First, it does not allow a comparison of the findings. Reasons for this can be found both in the objective circumstances of migration, i.e. factors related to the refugees' migration experience [10], and in the different methodological aspects and contexts of the studies reviewed. However, it is important to emphasize that the nature of data included in this review does not allow them to be decontextualized; it is the inclusion of specific contexts that makes these data meaningful in the first place. Second, the limitation concerns the use of human rights conceptual framework as a criterion for interpreting data and not as a criterion to collect scientific contributions. This choice was related to the relatively low dissemination of articles on this specific topic. Other studies must therefore be conducted to understand the role of mobile technologies in guaranteeing the application, enforceability and protection of asylum seekers and refugees fundamental human rights.

Although the "images of Syrian refugees holding mobile phones and capturing selfies upon reaching dry land have become potent symbols of migration in the 21st century" ([41], p. 58), and although metaphors of "connected migrants" [73] and "digital passages" [27] have affirmed the inexorable changes in the forced migration experience, many research areas are still unexplored.

Future studies should analyze in depth the relationship between mobile technologies, practices, and the needs of people on the move in specific contexts, such as contexts were vulnerable populations are considered. Furthermore, future studies should analyze the effects of transnational communication via mobiles on families and communities in the countries of origin, especially in terms of enhancing both social and identity capital growth. Studies in this review have tried to map out the ways in which refugees navigate the new digital communication infrastructures. Other studies should improve knowledge about how these infrastructures are navigated by government officials, human traffickers, corporations, and host population, too. Specifically, further studies should outline how the reliance on both infrastructures and artifacts could affect the interplay among the main protagonists of this "sociotechnical space" [27]: refugees, smugglers, corporations, and governments. Further studies should explore more directly the relationship between mobile phone use and the application, enforceability, and protection of asylum seekers and refugees' human rights.

## Supporting information

**S1 Appendix. Details of the 43 contributions reviewed.**
(DOCX)

**S2 Appendix. Prima ScR checklist.** Preferred Reporting Items for Systematic reviews and Meta-Analyses extension for Scoping Reviews (PRISMA-ScR) Checklist.
(DOCX)

## Author Contributions

**Conceptualization:** Tiziana Mancini.

**Data curation:** Tiziana Mancini, Federica Sibilla, Dimitris Argiropoulos, Michele Rossi, Marina Everri.

**Formal analysis:** Tiziana Mancini, Federica Sibilla, Dimitris Argiropoulos, Michele Rossi, Marina Everri.

**Funding acquisition:** Tiziana Mancini.

**Investigation:** Tiziana Mancini, Federica Sibilla, Marina Everri.

**Methodology:** Tiziana Mancini.

**Project administration:** Tiziana Mancini.

**Supervision:** Tiziana Mancini.

**Validation:** Tiziana Mancini.

**Writing – original draft:** Tiziana Mancini, Federica Sibilla, Dimitris Argiropoulos, Michele Rossi, Marina Everri.

**Writing – review & editing:** Tiziana Mancini, Federica Sibilla, Marina Everri.

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
