## [Decision Letter · Decision Letter 0]

9 Oct 2019

PONE-D-19-24667

The opportunities and risks of mobile phones for refugees’ human rights: A scoping review.

PLOS ONE

Dear Prof. Tiziana,

Thank you for submitting your manuscript to PLOS ONE. After careful consideration, we feel that it has merit but does not fully meet PLOS ONE’s publication criteria as it currently stands. Therefore, we invite you to submit a revised version of the manuscript that addresses the points raised during the review process.

Two reviewers have evaluated the manuscript. I agree with the overall positive comments and believe this contribution is worthy of consideration after moderate revisions will be provided. In particular, I agree with Reviewer 1 that the last sections of the review could be shortened/schematized a little for clarity. Also, I think the point made by Reviewer 2 on the appropriateness of the Human Rights framework is relevant and should be addressed with attention. 

We would appreciate receiving your revised manuscript by Nov 23 2019 11:59PM. To enhance the reproducibility of your results, we recommend that if applicable you deposit your laboratory protocols in protocols.io, where a protocol can be assigned its own identifier (DOI) such that it can be cited independently in the future. For instructions see: http://journals.plos.org/plosone/s/submission-guidelines#loc-laboratory-protocols

We look forward to receiving your revised manuscript.

Kind regards,

Stefano Triberti, Ph.D.

Academic Editor

PLOS ONE

Journal Requirements:

Reviewers' comments:

Reviewer's Responses to Questions

**Comments to the Author**

1. Is the manuscript technically sound, and do the data support the conclusions?

Reviewer #1: Yes

Reviewer #2: Partly

2. Has the statistical analysis been performed appropriately and rigorously? 

Reviewer #1: N/A

Reviewer #2: Yes

3. Have the authors made all data underlying the findings in their manuscript fully available?

Reviewer #1: Yes

Reviewer #2: Yes

4. Is the manuscript presented in an intelligible fashion and written in standard English?

Reviewer #1: Yes

Reviewer #2: Yes

5. Review Comments to the Author

Reviewer #1: This scoping review studies the literature on the role of mobile phones (MPs) in refugees’ experience in order to analyze the media practices in refugees’ everyday lives and if this use could be a risk or an opportunity for policies.

The manuscript is well organized ad well written. The authors present exhaustively the current state on this theme highlighting relevant aspects that already are known and gaps in the literature as well.

I think the methodology is well described and the discussion presents relevant points.

I have minor comments on content and methods (they follow), but I believe this review should be considered for publication.

- It is not clear why the systematic review flow (figure 1) presents authors in the search instead of the search enginges; moreover, PRISMA is mentioned but relevant methodological literature (Liberati, Moher, etc.) is not cited.

- Again on figure 1: the is something wrong in the numbers in the last phases ( 45 – 9 =37? …37+7=43? …probably the first one at full text screening is wrong); also, check the use of multiple character types in this figure (sometimes it is times new roman, sometimes calibri, etc.)

- The main weakness of the manuscript is that it is quite long and complex. I think this is acceptable giving the richness of the topic, however some parts could be shortened, for example final discussion sometimes appears repetitive with results… maybe, it is not necessary to describe all the articles one by one, but just relevant examples, then commenting relevant aspects in discussion with frequent reference to the table in appendixes

Furthermore: some schematic concepts could be turn into figures, e.g., the “five topics along a continuum” – this would improve readability and citability of the manuscript and shorten it a little

- I would correct “conclusion and discussion” in “discussion and conclusion”

- Last section could be shortened to focus on limitations (strengths have been made clear already in discussion in my opinion); moreover, authors sometimes merge limitations of the reviewed studies and limitations of their review itself, while only the latter should be mentioned here. In conclusion I think this section could be considerably shortened and just list possible methodological shortcomings and indications for future review efforts.

Reviewer #2: I found this scoping review to be of a high standard. In particular: The study presents the results of original research, or, in this case, of a systematic and novel review of the literature. It is informative, and should meet the growing scholarly demand for a review of the literature on migration and on digital media, with a focus on mobile phones (MPs). The database and bibliographic searches were performed to a high logical and technical standard, and exhaustively so, and are described in sufficient detail. The article is presented in an intelligible fashion, without unnecessary jargon and technicalities, and is written in standard English. To the best of this reviewer's knowledge, this bibliography-based review research meets all applicable standards for the ethics of research integrity. Likewise, the article adheres to appropriate reporting guidelines and community standards for data availability.

The article acknowledges the limitations of the scoping review, which is important. On p. 34, line 828, the authors acknowledge as a limitation that "the predominantly qualitative nature of the studies analyzed does not allow generalization of the findings". I (this reviewer) find that this conclusion might be problematic. The authors themselves acknowledge, as one of their crucial areas of consensus on the basis of their lit review, that results are "ambivalent", and that ultimately MPs provide risks and opportunities. This is not a limitation, therefore: it is a finding. Likewise, the fact that much of the data is contextual (p. 34, line834) is inescapable. The very nature of these data does not allow them to be decontextualized, and it is the inclusion of "contexts" that makes these data meaningful in the first place.

One limitation is more serious, in my opinion, but it is not addressed: I do not believe "human rights" to be the correct conceptual framework for this article and the literature it surveys. I would actually recommend that "human rights" be excluded from the very title, and replaced with tropes that are also used in the article, such as affordances, refugee experiences, etc.

The excruciatingly vast literature on human rights is not discussed at all here -- this is legitimate, as the article already accomplishes a lot. But if so, "human rights" cannot be presented as a crucial focus, as the title would imply.

Human rights are briefly discussed on p4, lines 84-86; p15, lines 334-337; p12, lines 260-261; p.25, line 596. In this reviewer's opinion, this is neither systematic/exhaustive nor particularly original, and sometimes it reads as an afterthought.

In light of this, the research question posed by the authors (p5, lines 109-112) is ultimately a necessary and important one, save for the implications re: human rights, which are not really shown as to being "guaranteed", "applied", "enforced", or "protected" on the basis of this review of migration and digital media/MPs. I would urge the authors to consider amending the research question, by excluding "human rights" and including "opportunities and risks of MPs in refugees' experience" (p.9, line 200), e.g., "empowerment", "personal agency", "integration", access to "services" (p.33). I believe this revision (which I would still consider fundamental, as it is a conceptual/analytical one, albeit easily feasible) to the conceptualization of the article, and to some of its phrasing, would make it more compelling, by better aligning the data/literature, the authors' conclusions, and the expectations of PLOS readers.

6. PLOS authors have the option to publish the peer review history of their article (what does this mean?). If published, this will include your full peer review and any attached files.

Reviewer #1: No

Reviewer #2: No

---

## [Author Response · Author response to Decision Letter 0]

9 Nov 2019

Dear Editor,

We want to thank the reviewers for the suggestions given. They have all been integrated into this version of the manuscript. Specifically, last sections (Discussion and Conclusion) of the review has been shortened/schematized a little for clarity and the point made by Reviewer 2 on the appropriateness of the Human Rights framework has been addressed with attention.

---

## [Editor Report · Decision Letter 1]

12 Nov 2019

The opportunities and risks of mobile phones for refugees’ experience: A scoping review.

PONE-D-19-24667R1

Dear Dr. Tiziana,

We are pleased to inform you that your manuscript has been judged scientifically suitable for publication and will be formally accepted for publication once it complies with all outstanding technical requirements.

With kind regards,

Stefano Triberti, Ph.D.

Academic Editor

PLOS ONE
---

## [Editor Report · Acceptance letter]

20 Nov 2019

PONE-D-19-24667R1 

The opportunities and risks of mobile phones for refugees’ experience: A scoping review. 

Dear Dr. Mancini:

I am pleased to inform you that your manuscript has been deemed suitable for publication in PLOS ONE. Congratulations! Your manuscript is now with our production department. 

With kind regards,

on behalf of

Dr. Stefano Triberti 

Academic Editor

PLOS ONE